# Facilitators and barriers to the implementation of a Mobile Health Wallet for pregnancy-related health care: A qualitative study of stakeholders' perceptions in Madagascar

Nadine Muller[1,2,3,4,5,6]*, Shannon A. McMahon[1,2,7], Jan-Walter De Neve[1,2], Alexej Funke[1,2], Till Bärnighausen[1,2,8,9], Elsa N. Rajemison[1,2], Etienne Lacroze[1,2], Julius V. Emmrich[4,5,6,10☯], Samuel Knauss[4,5,6,10☯]

1 Medical Faculty, Heidelberg Institute of Global Health, University of Heidelberg, Heidelberg, Germany, 2 University Hospital, University of Heidelberg, Heidelberg, Germany, 3 Department of Infectious Diseases and Pulmonary Medicine, Charité—Universitätsmedizin Berlin, Berlin, Germany, 4 Corporate Member of Freie Universität Berlin, Berlin, Germany, 5 Humboldt-Universität zu Berlin, Berlin, Germany, 6 Berlin Institute of Health, Berlin, Germany, 7 Department of International Health, Johns Hopkins Bloomberg School of Public Health, Baltimore, Maryland, United States of America, 8 Department of Global Health and Population, Harvard T.H. Chan School of Public Health, Boston, Massachusetts, United States of America, 9 Africa Health Research Institute, Somkhele and Durban, South Africa, 10 Department of Experimental Neurology and Center for Stroke Research, Charité—Universitätsmedizin Berlin, Berlin, Germany

☯ These authors contributed equally to this work.
* nadine.muller@charite.de

**Data Availability Statement:** As the datasets contain sensitive and potentially identifying

## Abstract

Financial barriers are a major obstacle to accessing maternal health care services in low-resource settings. In Madagascar, less than half of live births are attended by skilled health staff. Although mobile money-based savings and payment systems are often used to pay for a variety of services, including health care, data on the implications of a dedicated mobile money wallet restricted to health-related spending during pregnancy–a mobile health wallet (MHW)–are not well understood. In cooperation with the Madagascan Ministry of Health, this study aims to elicit the perceptions, experiences, and recommendations of key stakeholders in relation to a MHW amid a pilot study in 31 state-funded health care facilities. We conducted a two-stage qualitative study using semi-structured in-depth interviews with stakeholders ($N = 21$) representing the following groups: community representatives, health care providers, health officials and representatives from phone provider companies. Interviews were conducted in Atsimondrano and Renivohitra districts, between November and December of 2017. Data was coded thematically using inductive and deductive approaches, and found to align with a social ecological model. Key facilitators for successful implementation of the MHW, include (i) close collaboration with existing communal structures and (ii) creation of an incentive scheme to reward pregnant women to save. Key barriers to the application of the MHW in the study zone include (i) disruption of informal benefits for health care providers related to the current cash-based payment system, (ii) low mobile phone ownership, (iii) illiteracy among the target population, and (iv) failure of the MHW to

individual information, they are available to qualified researchers upon request from the Ethics Committee of Heidelberg University (Ethikkommission der Medizinischen Universität Heidelberg), Alte Glockengießerei 11/1, 69115 Heidelberg, Phone: +49 (0) 6221 562646-0.

**Funding:** Funding for the implementation of the Mobile Health Wallet was provided by grants from the Berlin Institute of Health and Else Kröner-Fresenius-Stiftung. TB was supported by the Alexander von Humboldt Foundation through the Alexander von Humboldt Professor award, funded by the Federal Ministry of Education and Research, Germany. The funders had no role in study design, data collection and analysis, decision to publish, or preparation of the manuscript. There was no additional funding received for this study.

**Competing interests:** The authors have declared that no competing interests exist.

**Abbreviations:** LMIC, Low and middle-income country; MDG, Millennium development goal; MM, Mobile money; MHW, Mobile health wallet; SEM, Social ecological model; SSA, sub-Saharan Africa.

overcome essential access barriers towards institutional health care services such as fear of unpredictable expenses. The MHW was perceived as a potential solution to reduce disparities in access to maternal health care. To ensure success of the MHW, direct demand-side and provider-side financial incentives merit consideration.

## Introduction

Health-related, out-of-pocket payments are a frequent source of impoverishment among households in low- and middle-income countries (LMICs) [1–3]. In its concept of universal health coverage within the 2030 Agenda for Sustainable Development, the World Health Organization emphasized a need to protect families against financial risks due to health care-related expenditures [4].

In several high-income countries, health insurance successfully protects against financial risks from health shocks based on the principle of solidarity: financial risk of illness is spread across the population by raising funds through prepayments or taxes and thereby pooling available funds. In LMICs, the implementation of similar concepts has been challenging due to a weakened health system, an absence of a centralized insurance platform and barriers in terms of household willingness and ability to join programs that require regular premium payments [5]. Alternative strategies such as community savings groups have been successful on a local level but are currently difficult to expand to a national scale [6].

The spread of mobile phones and mobile payment systems have changed the way economies and societies work in LMICs [7]. Today, more than 70% of worldwide mobile phone subscriptions come from LMICs with more than 74 subscriptions per 100 people in sub-Saharan Africa (SSA) in 2016 [8]. Following an exponential growth of mobile phone ownership in SSA in the last ten years, mobile payment systems, colloquially known as mobile money (MM), have disrupted the financing and banking landscape and have become a mainstay of financial transactions throughout SSA [9]. In 2017, three quarters of the world's MM transaction volume of almost 2 billion dollars were from SSA countries [10]. These technologies provide an unprecedented opportunity for inclusive solutions to the health coverage challenge.

### The mobile health wallet

Mobile-phone based payment and savings platforms aim to increase care-seeking among populations who do not have full coverage of essential health services. The system uses existing MM infrastructure and has the potential to restrict the withdrawal or spending of funds to the payment of services at registered health care facilities. A payment and savings tool like the MHW encourages households to financially prepare for future health expenses by securing their personal savings to better respond to health shocks. In case of imminent need, i.e. due to sudden illness or complications, such an approach further enables users to collect contributions from relatives or raise funds from other sources. Within recent years, MM based payment and savings platforms have been successfully applied in LMICs, including Kenya and Mali, for various medical conditions [11]. Pregnancy and birth-related care seems to be particularly well suited for the use of a MHW as expenses are mostly predictable both in time and amount. However, to our knowledge, there is currently no evidence on the success of application of a MHW specifically for maternal health care.

In Madagascar, the non-governmental organization *Doctors for Madagascar* developed a Mobile Health Wallet (MHW) to mitigate or eliminate out-of-pocket payments for health care

expenditures. The core system behind the MHW is a software platform that allows users, namely pregnant women, to save and pay for health care services using MM at participating health care providers. Personal data and funds on the MHW are linked to an individual SIM (subscriber identity module) card which can be used with any type of handset independent of smartphone capability or internet access. To further incentivize pregnant women to save money within their MHW, external donors provide bonus payments to women who reach a savings target [12]. Additional incentives include access to ambulance services, and obstetric ultrasounds at no cost based on need. To enable implementation of this digital payment system at health care facilities, tablets or mobile phones were distributed to the health care providers in participating centers.

In prior work, we found a high acceptability and perceived usefulness of a MHW to overcome financial obstacles to health care among Malagasy women [13]. The Madagascan Ministry of Health has also shown pointed interest in rolling out the MWH and, in cooperation with *Doctors for Madagascar*, is undertaking a pilot of the MHW in 31 state-funded health care facilities in the region of Analamanga in central Madagascar from which data for this research is drawn.

## Maternal and neonatal health care in Madagascar

Assuring the accessibility and acceptability of health services for pregnant women is a persistent challenge for Madagascar. Key millennium development goal (MDG) 5 indicators for maternal health were not met: despite falling gradually to 440 deaths per 100,000 livebirths in 2015, the maternal mortality rate was far short of the MDG target of 190 per 100,000 [14,15]. The latest available data from 2017 show a national maternal mortality rate of 335 deaths per 100,000 live births [16]. According to the most recent data from 2009 and 2013, the percentage of deliveries attended by skilled health personnel has stagnated at 44% [17]. Although there is no universal health coverage, Madagascan national health policy outlines the free provision of antenatal care and for normal, non-complicated deliveries in all public-sector health care facilities. Fees are often charged for folic acid and iron supplements, lab tests, drugs and related treatment as well as complications during delivery, which effectively limits women's access to skilled maternal care [13,18]. As previously shown, a majority of pregnant women in Madagascar use household cash savings to financially prepare for institutional delivery. However, among poorer populations it is especially difficult to save cash while also saving for expenses such as school fees or unforeseen financial crises. This situation thus threatens the feasibility of saving adequate funds for maternal health services such as antenatal care and skilled birth attendance [13].

## A study of stakeholders' perceptions

For health care professionals, the implementation of an alternative payment modality, such as the MHW, inevitably entails alterations in clinical routines and revenue streams. These alterations have the potential to impact stakeholders' administrative and organizational habits and alter economic benefits inherent to existing payment systems. Adaptation to the regional contexts, understanding stakeholders' perspectives about usefulness, practicability and acceptability is therefore paramount for efficient project implementation and scale-up [19–21]. This qualitative study explores stakeholders' perspectives regarding the proposed implementation of the MHW by learning how stakeholders describe obstacles in providing pregnancy and delivery care in relation to the MHW, and to analyze potential pitfalls and recommendations prior to introducing the MHW in public-sector health care facilities.

## Materials and methods

### Ethical considerations

The study was approved by the Madagascan Ministry of Health and the Heidelberg University Hospital Ethics Committee (No. S-703/2017). Informed written consent was obtained from all participants of the study prior to the interview.

### Study setting

The study was conducted in Madagascar, an island country with 26.3 million inhabitants off the coast of East Africa [22]. According to the Human Development Index, Madagascar ranks 161[th] of 189 countries [23]. Annual income per person is low at US$461, placing Madagascar among the 6 poorest countries globally in 2018 [24]. Poverty is especially prevalent in rural areas where around two-thirds of the population live [25,26]. As of 2018, 41% of the population are below age 14. The government expenditures on education measure 2.1% of the gross domestic product (GDP); an estimated 3.93 million adults are illiterate [27]. Educational deprivation strongly varies across socioeconomic groups: the richest quintile of the population averages 9.8 years of schooling, compared with 1.7 years for the poorest quintile [23].

Located in Central Madagascar's Analamanga region, the study districts (Atsimondrano and Renivohitra), which include the capital Antananarivo, are primarily urban and were selected due to an extensive phone network coverage and a high population density. Pregnancy and birth related health care services in this area are predominantly provided by Centres de Santé de Base level 1 (CSB1), equivalent to health posts in anglophone African countries, and CSB level 2 (CSB2), equivalent to health centers and district hospitals [28].

### Study sampling

We conducted a two-staged qualitative study using semi-structured interviews. In the first stage, we purposively sampled sector experts across health care system levels based on their experience and ability to provide relevant information on health care organization, provision, communication, administration and financing in the study zone [29]. We complemented this approach in a second stage with snowball sampling and ultimately interviewed respondents across different health care and health profession levels, community representatives and telephone provider representatives [30,31].

### Data collection

Semi-structured interview guides were developed in English and translated into French and Malagasy. Data were collected from 28[th] of November until 12[th] of December of 2017 by interviewers fluent in Malagasy, English and French, and with graduate-level education and experience in qualitative research methods. All members of the data collection team participated in a one-day workshop and training to ensure a common understanding of all aspects of the MHW intervention. Prior to the start of interviews, we collected basic socio-demographic data of respondents (gender, age, years of experience in the health sector). Overall, 21 stakeholders were sampled including 8 community representatives, 8 health care providers, 3 health officials and 2 phone provider representatives. All in-depth interviews were audio-recorded in a private place of the respondent's choice. Interviews lasted 25 to 75 minutes. To assure a clear understanding of the concept and aim of the MHW by respondents, a concise description of the MHW was read aloud and interviewees were encouraged to ask questions and seek clarity throughout the interview. Figs 1 and 2 depict the study sample and study themes addressed across interviews, respectively. Topics included general perceptions of the MHW, sector and

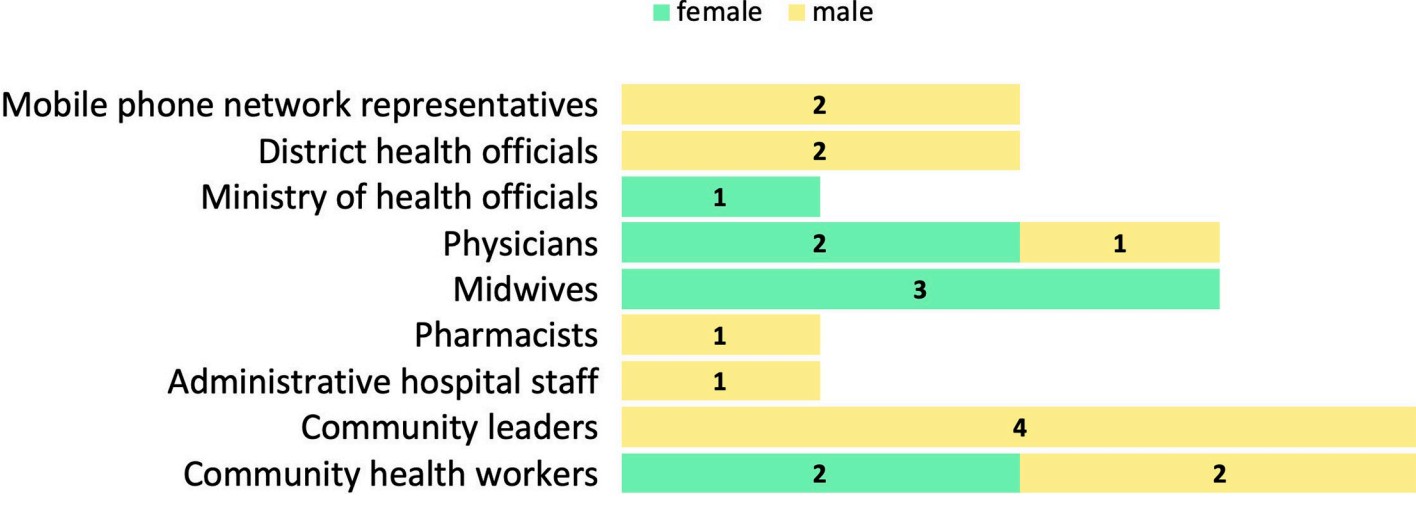

**Fig 1. Number of participants among groups and gender distribution (N = 21).**

component experiences, perceptions of obstacles to providing health care, as well as recommendations and expectations for implementing MHWs. Sampling continued until no new information emerged and saturation was reached [32].

## Data analysis

All interviews were tape-recorded, verbatim transcribed in the original language and, if in French or Malagasy, translated into English. Transcripts were analyzed by two researchers (NM, AF) who developed a codebook which was validated by a senior qualitative research expert (SM) [30]. Each set of data was analyzed at least twice. Thematic coding was applied by using a partially deductive approach, informed by findings from a previous study, and an inductive coding technique with creation of themes as they emerged from the transcripts [13].

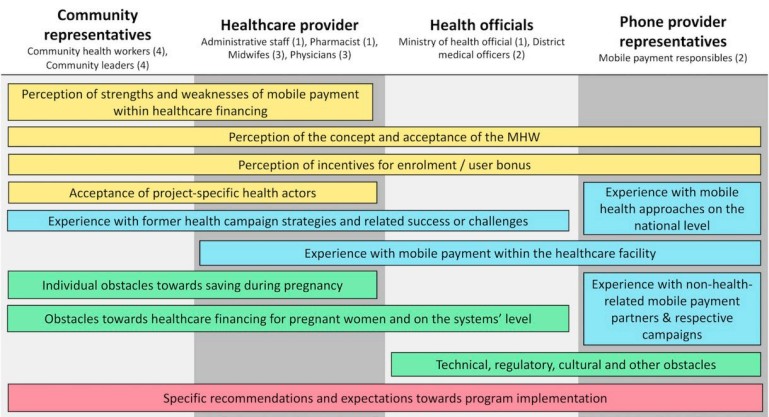

**Fig 2. Overview of respondent characteristics by level, type, number (n) and topics covered (N = 21).**
Yellow = perception of MHW components, Blue = sector and component experiences, Green = obstacles towards MHW components, Red = recommendations and expectations.

Analysts convened regularly to debate differences in coding and interpretation. During the analysis process, when new codes emerged, coders discussed the new findings and transcripts were recoded as the codebook was refined.

In the process of coding, a social ecological model (SEM) emerged as an organizing framework around which to present the data. The SEM is a theory-based framework that considers the interplay of multiple social system levels and interactions from the individual level to the broader environment [33]. The SEM facilitated the exploration of dynamic interactions across different social levels that influence health care seeking behaviors and program acceptance. Emerging themes and subthemes with potential influence on successful MHW implementation were grouped into the following three SEM layers and are presented accordingly in the results section:

i. *Institutional and policy*: Influences of the current health care system and implications of national and local laws and policies. Sub-themes include previous experience with mobile payment schemes, implications by the existing service payment system, potential conflicts with external service providers and legal and technical factors with implications on the MHW.

ii. *Interpersonal and community*: Factors by formal and informal social networks and social support systems. Subthemes describe interpersonal and cultural influences on decision-making, influences by applied saving methods and factors related to communication strategies.

iii. *Individual*: Factors and characteristics of the individual that influence behavior, expectations and attitude towards the MHW. Subthemes focus on the minimum prerequisites for access and factors influencing the individuals' decision-making.

Coding was performed with the software NVivo 12 [34].

## Results

Fig 3 presents a summary of the influencing factors on the implementation of the MHW as identified during the analysis process.

### Institutional factors

All interview respondents describe mobile technologies as well known and having become very common for a multitude of daily life activities. However, only one respondent had previous experience with mobile payment systems within the health care sector. In this specific case, a non-governmental organization provided program-specific mobile payments within the Malagasy health care sector.

MHW-induced changes of the current reward systems might arouse opposition and concerns. One phone provider representative recalled a particular example of the implementation of a novel mobile payment system within the energy sector wherein the introduction of the new system induced the replacement of former reward and payment structures and ultimately changed official or non-official benefits of some actors. This led to frustration and in consequence a disparaging or even hostile attitude toward the program.

As a relevant obstacle to health care generally but also an important factor that could endanger acceptance of the MHW, interviewees mentioned inconsistent service fees at the provider level. This point was primarily raised by the health care providers themselves. A physician stated:

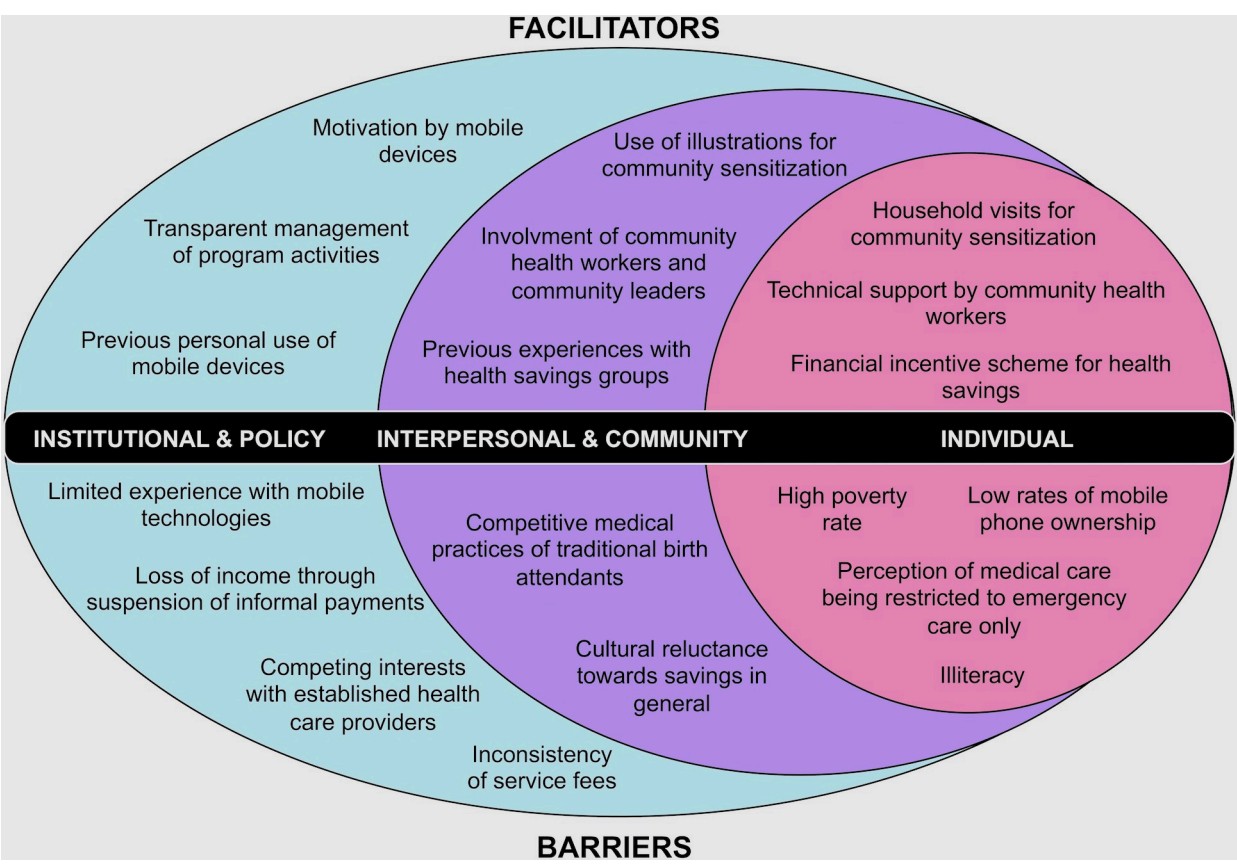

**Fig 3. Social ecological model with layers of influence and influencing factors on the implementation of the MHW.**

"*Not knowing the price for delivery is an obstacle that deters (pregnant women) from going to hospital. Because in their mind, it is about money and it costs a lot to give birth at a hospital. This is a big problem, a real obstacle for them.*" (*Female physician, state-funded health care facility*)

Especially non-official stakeholders were particularly concerned about potential non-transparent management of program activities including unclear definitions of MHW benefits and complicated terms and conditions. Respondents also described how scarce personal savings for birth could induce hesitancy towards saving activity with the MHW, due to its restricted pay-out options. This can be especially critical for poorer women with no additional cash funds available to react in case of other emergencies. Thus, clear and comprehensive program information for all program participants was deemed essential to avoid negative impacts on women's participation.

Other concerns were voiced specifically towards some MHW's components such as the ambulance service and mobile ultrasound free of charge. Conflicts with existing care providers, offering identical examinations as a paid service, were predicted. In order to avoid competition, collaboration with the existing provider for execution of program activities was cited as a potential solution.

The planned distribution of a mobile device (tablet or mobile phone) for program specific purposes to each participating health care unit was seen as highly beneficial by health care providers. They noted that this could be perceived as a "bonus", and could thus promote

participation, and a positive attitude in general towards the intervention. One respondent asked whether the tablet could also be used for personal activities.

From a legal and a technical point of view, neither health officials nor phone provider representatives voiced concerns regarding the implementation of the MHW within state-funded health care institutions.

### Interpersonal and community factors

Decisions about antenatal care services and enrolment in program activities such as the MHW were not perceived as fundamentally shaped by the influence of husbands, family members or close relatives but rather by matrones. Matrones are traditional birth attendants who offer services, often at a lower price, as an unofficial alternative to the public sector. Given their status, lower service prices and the ability to sway women's decisions, matrones were described as crucial in terms of decision making for especially poorer women.

Few respondents reported knowing women who use a mobile money account for financial birth preparation and only a minority reported experience with savings on mobile wallets in general. Another very recent savings method mentioned by respondents involves the concept of a rotating savings account (locally known as *akandray akandray*) where several women regroup for a common saving box, which is available for one particular woman in times of need. As reported by the respondents, as of yet, use of mobile currency has not been explored for rotating savings accounts in Madagascar, which have relied on cash savings thus far.

As mentioned independently by phone provider representatives, an obstacle to financial birth preparedness was described as a cultural reluctance to save money, with the exception of the personal savings box *(caisse noire)*. A phone provider representative familiar with the implementation of MM services cited reluctance as a reaction to environmental factors and general distrust in financial affairs among the population:

> "*I think saving has always been problematic in [a low-income country like] Madagascar because of its characteristics in general: You are entrusting your money into somebody else's hand. That is something that may scare people because they may think that they will not have access to their funds anytime they feel they want them.*" (Male phone provider representative (Malagasy native))

Some community leaders attributed this reticence toward saving to a discomfort or fear when entering official or financial institutions, where lower-income men and women feel unwelcome or unaccepted. Overall, stakeholders explicitly agreed that the use of a mobile payment system within the health care sector could bypass this barrier:

> "*With this (MHW), she is saving herself money in the telephone, so it solves the issue of going to a bank. Because this is a true fact, we Malagasy are afraid of going to any office or bank. So it solves a big problem.*" (Male community leader)

Actively involving existing structures of community health workers into the programs' sensitization campaign was strongly recommended, because these individuals were trusted in local communities and associated with other programs related to health and community betterment. To be distinguished from former campaigns, respondents recommended a communication design with clear and unambiguous messages and a heavy reliance on illustrations rather than text.

Cited as a crucial element of success, all respondents deemed a close collaboration with community leaders (locally known as chef *fokontany*) as essential for project acceptance among the population.

"*If you want to work with (populations from a lower social strata), you should always work with the head (meaning here: community leader, locally known as chefs fokontany). Because we are seen as parents and we are listened to. If you don't follow this advice but come with an own initiative–this will for sure be negative.*" (Male community leader)

## Individual factors

Respondents perceived saving with the aid of a dedicated mobile account as a desirable-yet-challenging aspect of the MHW. Most stakeholders, especially community health workers, estimated that pregnant women's scarcities of funds (i.e. due to lacking job opportunities, absence of regular income, financial burden for daily expenses, opportunity costs in the study zone) would serve as major obstacles to successful application of the MHW's savings function:

"*The small amount of money earned today is spent for the food today. And the price of rice is currently very high and even increasing every day. So, it is impossible for them (pregnant women) to save because their income is not regular and stems from a precarious job.*" (Female community agent)

Most stakeholders said that non-monetary incentives such as supplements, ultrasound testing or free ambulance transfer would not effectively promote the use of the MHW because they do not tackle the principle financial challenges of the target group. As the only means to effectively overcome these financial barriers, the implementation of direct financial incentives rewarding pregnant women for even minor saving activity were strongly encouraged by all stakeholder groups.

As a second major obstacle, respondents described limited mobile phone ownership among poorer women, which would essentially exclude an estimated majority of women–particularly those from lower population strata—from participation. The estimates of phone ownership rates, however, were contradictory across respondent groups with community leaders and health workers raising concerns and health officials being more optimistic about mobile phone ownership rates.

Illiteracy among poorer women was also mentioned as a potential barrier limiting women's abilities to directly follow program information and advice distributed via text-based mobile phone messages. Respondents said the availability of formalized, individual support personnel could improve user experiences and avoid dependency on others for technical support which could make women susceptible to fraud.

Stakeholders also said that a limited education level among some women would be a barrier to seeking professional maternal health care. Similarly, respondents mentioned that some women have incorrect medical knowledge or a perception of non-usefulness of professional medical care unless complications occur.

Respondents further described the MHW as incapable of fully mitigating several underlying barriers such as fear of unpredictable expenses at hospitals and related expenses for transportations and provisions, leading some women to forgo birth attendance in qualified centers:

"*They (pregnant women) don't dare approaching health centers, because they think that entering health centers always mean 'money out'. Self-medication or consultation of local matrones (traditional birth attendants) therefore dominate.*" (Female community agent)

"*Even if they come to the health center for antenatal care, some of them are still giving birth at home because the matrone is less expensive compared to the hospital. And they want to avoid additional expenses like buying drugs or food.*" (Male community agent)

For program sensitization purposes, some respondents recommended confronting the target group with radical or even threatening scenarios. One community leader suggested explicitly conveying the risk for death when facing complications without attending skilled care. For program information purposes, respondents from within the community advised approaching the target population directly within their living area and undertaking home visits.

*"I don't have any other advice than getting out into the communities directly. It is not efficient to call them to come to the CSB (centre de santé de base, = basic health facility). The problem nowadays is time; time is precious. You will not get anyone spending easily their time. We, here on the ground, we can see that. There is no better alternative than going door to door and seeing here one woman, there another one."* (Male community leader)

## Discussion

We elicited facilitators, barriers and recommendations for the implementation of a mobile phone-based savings and payment platform in Madagascar from local stakeholders. All stakeholders confirmed usefulness and appropriateness of the MHW. Low mobile phone ownership, general factors restricting acceptance of institutional health care, cultural reluctance and inexperience with saving, illiteracy and potential pushbacks from health care providers due to suspension of status quo's benefits were seen as critical challenges. A result-based bonus system for providers and users, the design of promotion and outreach activities and the respect and inclusion of local power hierarchies during program implementation were seen as key determinants of the MHW's success.

Formative research is used to understand attributes of the target audience and the context in which interventions will be designed [35]. Ideally, formative research is conducted before the setup and implementation of an intervention. Several publications exist that draw upon formative research to adapt and enhance the design of mobile phone-based interventions to support maternal health care in LMICs [36–39]. Although mobile phone-based payments schemes are changing the landscape of health care provision in other SSA countries as described in our previous work [13], to our knowledge, publications on formative research regarding the implementation of a mobile payment and savings system within health care in LMIC are scarce.

Most respondents estimate that financial incentives, i.e. by bonus payments to pregnant women, are needed to foster curiosity and curb hesitation regarding MHWs. With growing evidence that conditional cash transfers have the potential to increase women's access to maternal health care, demand-side payment schemes are becoming increasingly popular in LMICs [40–43]. It has also been shown that conditional cash transfers contribute to reduced time taken before presenting for emergency obstetric care, thus reducing maternal morbidity and death [44]. Stakeholders strongly recommended the introduction of a bonus-payment scheme in order to promote women's saving activities and to facilitate access to skilled maternal care. This recommendation was argued first on the grounds of equity in health care access and second in relation to the success of the intervention.

Stakeholders' estimates on mobile phone ownership and literacy among pregnant women gave mixed results. Within our previous study, we found more than three quarters of the participants have access to a mobile phone and more than nine out of ten women are literate [13]. These findings are contradictory to the estimates of the stakeholders in the present study and reflect a point of misunderstanding that merits consideration. An essential limitation of the previous study was that only visitors of a health care center were interviewed, which suggests a selection bias as those who could not seek care were excluded from the study. Based on these

contradictory but potentially fundamental findings and the prevailing reality of adult illiteracy in the Malagasy population, we strongly suggest applying an image-based knowledge dissemination strategy and alternatively offering verbal announcements and voice calls to convey basic program information. Furthermore, we recommend evaluating whether several saving accounts could be established on one mobile phone to potentially allow sharing one phone among several family members or among a group of women to bridge a potential low phone ownership rate. However, it needs to be considered that a single SIM card for several people raises concerns of confidentiality and security of individual savings. We therefore suggest making SIM cards largely available to pregnant women even without owning a phone as they can be inserted and removed easily with any handset, thus mitigating the aforementioned privacy concerns.

Our findings reflect a number of stakeholders' recommendations and challenges towards a successful implementation of the MHW. One of the strongest suggestions pronounced by all stakeholder groups was to respect local hierarchies and to opt for intense cooperation with local leaders. A mixed-methods study on lessons learnt from the implementation of a maternal health care voucher scheme in Eastern Uganda describes how communities were more inclined to listen and trust messages from their local leaders, thus leading to enhanced project participation [45]. Broad communication seems to be an equally important respondent-driven component of MHW acceptance. Our results suggest that promotion of the health care campaign should make use of various media channel, yet place particular emphasis on personal contact. Similar to our findings, a research team from Cameroon found that social mobilization activities such as door-to-door visits and involvement of mass media represented the most important and accepted ways of communicating campaign information [46].

Within this formative research, we experienced no principle resistance to change as it has been described for other mobile health interventions [47]. Potential challenges for the health care provider might be changes in habitual work processes and higher workload. Attenuation of these events and avoidance of negative consequences should be ultimately foreseen within the program intervention plan [48]. Cessation of former benefits could lead to reduced motivation or even hostile behavior towards new interventions. Therefore, a balance of benefits and incentives of participation should be carefully evaluated for each stakeholder group to avoid or replace a perceived loss of current benefits. Among interviewed health care providers, there was great interest in the mobile tablet distributed for implementation of the intervention to participating health care facilities with particular interest in personal usage of these devices. A study seeking to improve maternal health care service delivery in Ethiopia found that entrusting a mobile device to health care staff creates a sense of ownership and has the potential to be a strong motivator to undertake proposed intervention activities [49]. The provided mobile devices could be purposely unblocked so that personal calls, messages and internet consumption were available for users. It should be discussed whether this form of incentive should be applied to the MHW intervention.

Our study has several limitations. The study was conducted in an urban setting, and expectations or experiences in rural areas likely differ. Additionally, the study was conducted in preparation for a specific health care support program. This could have led to a social desirability bias as participants may have wanted to appeal to the interviewers in order to be selected for pilot implementation of the MHW activities. A cultural aversion to saving was described as a major concern by several stakeholders, but this is not reflected in the literature, nor in our previous study among 412 members of the target population, who overall reported a high willingness to save [13]. Given the divergence between community members' self-reports versus high-level stakeholder perceptions of communities, the relevance of sideways data from stakeholders in this specific context possibly needs to be considered with caution.

Overall, in lack of insurance schemes, saving funds help households and individuals to withstand financial shocks. There is clear evidence that poorer populations have substantial demand for savings despite various constraints [50,51]. Although the MHW's primary goal is to reduce barriers to health care services for populations most in need, women from the lower socioeconomic strata, including women who lack basic school education and/or the means to afford a mobile phone or SIM card, might not sign up to the MHW. To minimize health inequities, we strongly recommend more research on prevailing prerequisites of access, leading to a more inclusive design of the MHW.

## Conclusions

Respondents in this study described the implementation of a mobile phone-based savings and payment platform as a useful and potentially powerful means to reduce inequalities in access to skilled maternal care. The design of demand-side and provider-side benefits in the form of bonus payments were viewed as important determinants underpinning the success or failure of the MHW. Stakeholders also emphasized the need for a culturally sensitive information and promotion campaign for intervention success. For success, stakeholders and local leaders must know of and actively support the MHW program. More research is needed to address factors determining access to health care services among pregnant women from lower socioeconomic strata in more detail.

## Supporting information

**S1 Appendix. Questions asked to participating stakeholders from different levels.** (PDF)

## Acknowledgments

The authors thank the Madagascan Ministry of Health and all stakeholders who provided their support for this study. The authors thank Mialy Rakontondraina for critical revision of the data collection tools and Dr. Jeannot Randriantsoa for his support during data collection.

## Author Contributions

**Conceptualization:** Nadine Muller, Shannon A. McMahon, Julius V. Emmrich, Samuel Knauss.

**Data curation:** Nadine Muller.

**Formal analysis:** Nadine Muller, Shannon A. McMahon.

**Funding acquisition:** Nadine Muller, Julius V. Emmrich, Samuel Knauss.

**Investigation:** Nadine Muller, Alexej Funke, Elsa N. Rajemison, Julius V. Emmrich, Samuel Knauss.

**Methodology:** Nadine Muller, Shannon A. McMahon, Jan-Walter De Neve.

**Project administration:** Elsa N. Rajemison, Julius V. Emmrich, Samuel Knauss.

**Supervision:** Shannon A. McMahon, Jan-Walter De Neve, Till Bärnighausen, Julius V. Emmrich, Samuel Knauss.

**Validation:** Shannon A. McMahon, Jan-Walter De Neve, Alexej Funke, Till Bärnighausen, Elsa N. Rajemison, Etienne Lacroze.

**Visualization:** Nadine Muller.

**Writing – original draft:** Nadine Muller.

**Writing – review & editing:** Nadine Muller, Shannon A. McMahon, Jan-Walter De Neve, Till Bärnighausen, Etienne Lacroze, Julius V. Emmrich, Samuel Knauss.

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
