## [Decision Letter · Decision Letter 0]

18 Nov 2019

PONE-D-19-28907

Facilitators and barriers to the implementation of a Mobile Health Wallet for pregnancy-related health care: a qualitative study of stakeholders’ perceptions in Madagascar

PLOS ONE

Dear Nadine Muller ,

Thank you for submitting your manuscript to PLOS ONE. After careful consideration, we feel that it has merit but does not fully meet PLOS ONE’s publication criteria as it currently stands. Therefore, we invite you to submit a revised version of the manuscript that addresses the points raised during the review process.

We would appreciate receiving your revised manuscript by 20th December. To enhance the reproducibility of your results, we recommend that if applicable you deposit your laboratory protocols in protocols.io, where a protocol can be assigned its own identifier (DOI) such that it can be cited independently in the future. For instructions see: http://journals.plos.org/plosone/s/submission-guidelines#loc-laboratory-protocols

We look forward to receiving your revised manuscript.

Kind regards,

Sharon Mary Brownie

Academic Editor

PLOS ONE

Journal Requirements:

"Funding for the implementation of the Mobile Health Wallet was partly provided by a grant from the Else Kröner-Fresenius-Stiftung. TB was supported by the Alexander von Humboldt Foundation through the Alexander von Humboldt Professor award, funded by the Federal Ministry of Education and Research, Germany. The funders had no role in study design, data collection and analysis, decision to publish, or preparation of the manuscript.".

i) Please provide an amended statement that declares *all* the funding or sources of support (whether external or internal to your organization) received during this study, as detailed online in our guide for authors at http://journals.plos.org/plosone/s/submit-now.  Please also include the statement “There was no additional external funding received for this study.” in your updated Funding Statement.

ii) Please include your amended Funding Statement within your cover letter. We will change the online submission form on your behalf.

Additional Editor Comments (if provided):

This is an interesting study, however, reviewers have identified some major issues which you will need to address. First, the abstract and manuscript must also be formatted correctly to the journal requirements. Other feedback includes the requirement of a much deeper analysis of the data with more participant comments to justify conclusions. Please look very carefully at each requirement tabled by the reviewers and address each of these in a table of response along with your revised manuscript.

Reviewers' comments:

Reviewer's Responses to Questions

**Comments to the Author**

1. Is the manuscript technically sound, and do the data support the conclusions?

Reviewer #1: Yes

Reviewer #2: Yes

2. Has the statistical analysis been performed appropriately and rigorously? 

Reviewer #1: N/A

Reviewer #2: N/A

3. Have the authors made all data underlying the findings in their manuscript fully available?

Reviewer #1: Yes

Reviewer #2: No

4. Is the manuscript presented in an intelligible fashion and written in standard English?

Reviewer #1: Yes

Reviewer #2: Yes

5. Review Comments to the Author

Reviewer #1: This was an interesting study exploring one of the useful ways fintech could be used to achieve the SDGs in sub-Saharan Africa.

There are a few comments for the authors to address:

1. The abstract needs to be structured and in line with the journal guidelines.

2. Page 4, paragraph 1, line 86 needs to be revised for typo

3. Page 6, paragraph 1, line 133 needs revising. I think "include" should be substituted with the word "including"

4. On page 7, paragraph 2, line 170, the sentence "Gender distribution of participations as follows" needs to be revised.

5. The quotation marks in the quotations need to be revisited.

6. Can the authors provide more quotations from the participants especially for the information provided on page 13?

Reviewer #2: What is the current maternal mortality rate of Madagascar? Has it increased or decreased between 2015 and now? Beyond ante natal care what happens? Is delivery at a hospital or clinic free? Only the ante natal care was discussed. What is the socio economic status and literacy rate of the population in Madagascar and the catchment area where the study was done.

Line 109-110 use the latest indicators/statistics.

For me the paper fails to connect the dots between pregnancy health related care and the MHW?

Line 146 Prior to interview ---- move this to data collection, does not fall under sampling, unless if the data were used to decide who could be interviewed.

The data analysis needs to be explained in more detail for the reader to understand how the themes were arrived at and to illustrate an audit trail underpinning the trustworthiness. I like the way the conceptual framework was weaved into the findings. However, it would be good to mention the sub-themes within the broader categories of the conceptual framework

Line 170 - gender and distribution - can this be put in a table format? it would be easier to read in a table format.

Discussion; The discussion should highlight the limitations of the MHW given the low socioeconomic status of most of the pregnant mothers as well as the absence of handsets and the literacy rates. The illiteracy was mitigated well by suggestions of use of illustrations for health education. Or perhaps given these findings this is something that should be tried with a middle income cohort of pregnant women?

The suggestion of several people using a handset should take into consideration issues of confidentiality. I wish this question had been posed to the participants to hear their take on it. In my experience, many men would not want to share a phone with a woman.

This was an interesting study that puts a different spin to mHealth and health insurance, as the women would be in control of the money, not the health insurance.

6. PLOS authors have the option to publish the peer review history of their article (what does this mean?). If published, this will include your full peer review and any attached files.

Reviewer #1: No

Reviewer #2: No

---

## [Author Response · Author response to Decision Letter 0]

30 Dec 2019

We would like to first thank the reviewers for their helpful and constructive feedback. We are grateful to Reviewer 1 who described our work as one useful step towards achievement of the Sustainable Development Goals. We are thankful to Reviewer 2 who noted the empowering aspect of the digital intervention we investigate. Finally, we thank the editorial team for highlighting the interesting and sound character of our study. 

Please see below a point-by-point response to the comments from each reviewer. We are hopeful that these changes (also highlighted via track changes in the attached manuscript) meet the expectations of reviewers and editors. We look forward to your feedback.

1. The abstract needs to be structured and in line with the journal guidelines.

-> We thank the reviewer for highlighting this. We have followed the format of PLOS ONE author guidelines (which discourages the use of sub-headings in abstracts). The abstract has been formatted according the journal’s most recent author formatting guidelines available online at: (https://journals.plos.org/plosone/s/file?id=wjVg/PLOSOne_formatting_sample_main_body.pdf). 

To improve the overall clarity of the abstract we rephrased some of the wording as follows (page 2, line 33):

In cooperation with the Madagascan Ministry of Health, this study aims to elicit the perceptions, experiences, and recommendations of key stakeholders in relation to a MHW amid a pilot study in 31 state-funded health care facilities.

2. Page 4, paragraph 1, line 86 needs to be revised for a typo.

-> We rectified the typo accordingly:

… the payment of services…(page 4, line 81)

3. Page 6, paragraph 1, line 133 needs revising. I think "include" should be substituted with the word "including"

-> The wording has been revised and adapted: 

Located in Central Madagascar’s Analamanga region, the study districts (Atsimondrano and Renivohitra), which include the capital Antananarivo, are primarily urban and were selected due to an extensive phone network coverage and a high population density. (page 7, line 157) 

4. On page 7, paragraph 2, line 170, the sentence "Gender distribution of participations as follows" needs to be revised.

-> We have extracted the gender distribution from the legend and created an extra figure (Fig 1, page 8, line 192) to clarify the gender distribution among different groups of participants. 

5. The quotation marks in the quotations need to be revisited.

-> We revised and standardized the quotations marks (pages 10, 12 and 13) according to the journal’s guidelines. 

6. Can the authors provide more quotations from the participants especially for the information provided on page 13?

-> We thank the reviewer for this suggestion. By revising the analyzed data, we extracted two more quotations on unpredictable expenses and avoidance of care-related expenses. The following quotations have been added (page 15, line 362): 

“They (pregnant women) don’t dare approaching health centers, because they think that entering health centers always mean ‘money out’. Self-medication or consultation of local matrones (traditional birth attendants) therefore dominate.” (Female community agent)

“Even if they come to the health center for antenatal care, some of them are still giving birth at home because the matrone is less expensive compared to the hospital. And they want to avoid additional expenses like buying drugs or food.” (Male community agent)

1. What is the current maternal mortality rate of Madagascar? Has it increased or decreased between 2015 and now? Beyond ante natal care what happens? Is delivery at a hospital or clinic free? Only the ante natal care was discussed. 

2. What is the socioeconomic status and literacy rate of the population in Madagascar and the catchment area where the study was done?

-> We thank the reviewer for pointing out this important, but thus far missing information. To provide more context, enhance clarity, comprehensiveness and timeliness, we have added information on the maternal mortality rate and skilled birth attendance as suggested (page 5, line 112): 

Key millennium development goal (MDG) 5 indicators for maternal health were not met: despite falling gradually to 440 deaths per 100,000 livebirths in 2015, the maternal mortality rate was far short of the MDG target of 190 per 100,000 (14,15). The latest available data from 2017 show a national maternal mortality rate of 335 deaths per 100,000 live births (16). According to the most recent data from 2009 and 2013, the percentage of deliveries attended by skilled health personnel has stagnated at 44% (17).

Although the national health policy includes free provision of antenatal care and delivery in public-sector facilities, fees are charged for all drugs, additional tests and treatments. We specified that this is also the case for complications during delivery. We furthermore amended the following paragraph to clarify that financial obstacles substantially hinder access to skilled care in the current local context (page 5, line 118): 

Although there is no universal health coverage, Madagascan national health policy outlines the free provision of antenatal care and for normal, non-complicated deliveries in all public-sector health care facilities. Fees are often charged for folic acid and iron supplements, lab tests, drugs and related treatment as well as complications during delivery, which effectively limits women’s access to skilled maternal care (13,18). As previously shown, a majority of pregnant women in Madagascar use household cash savings to financially prepare for institutional delivery. However, among poorer populations it is especially difficult to save cash while also saving for expenses such as school fees or unforeseen financial crises. This situation thus threatens the feasibility of saving adequate funds for maternal health services such as antenatal care and skilled birth attendance (13). 

In the description of the study setting (page 6, line 149), we also added information on socioeconomic status and illiteracy in the country and in the catchment area: 

According to the Human Development Index, Madagascar ranks 161th of 189 countries (23). Annual income per person is low at US$461, placing Madagascar among the 6 poorest countries globally in 2018 (24). Poverty is especially prevalent in rural areas where around two-thirds of the population live (25,26). As of 2018, 41% of the population are below age 14. The government expenditures on education measure 2.1% of the gross domestic product (GDP); an estimated 3.93 million adults are illiterate (27). Educational deprivation strongly varies across socioeconomic groups: the richest quintile of the population averages 9.8 years of schooling, compared with 1.7 years for the poorest quintile (23). 

3. Line 109-110 use the latest indicators/statistics.

-> We updated our literature and database search to find more current databases and publications (including but not limited to data from UNICEF, WHO, DHS World Bank as well as from the Malagasy National Statistical Institute). Unfortunately, all agency and inter-agency databases containing information about skilled birth attendance in Madagascar relate to data not more recent than 2013. 

4. For me the paper fails to connect the dots between pregnancy health related care and the MHW?

-> We thank the reviewer for highlighting our shortcoming to explain which essential gap in accessing pregnancy health related care will be filled by the MHW in more clarity. In addition to the amendments we made in response to the reviewer’s first comment, we further highlighted the fact that charged fees for supplements and tests are an important, financial barrier to skilled maternal health care. 

We added some additional explanations on the potential risks faced with cash savings (page 5, line 122): 

As previously shown, a majority of pregnant women in Madagascar use household cash savings to financially prepare for institutional delivery. However, among poorer populations it is especially difficult to save cash while also saving for expenses such as school fees or unforeseen financial crises. This situation thus threatens the feasibility of saving adequate funds for maternal health services such as antenatal care and skilled birth attendance (13).

Within the description of the MHW (section: The Mobile Health Wallet, page 4, line 81), we clarified potential benefits of the MHW: 

A payment and savings tool like the MHW encourages households to financially prepare for future health expenses by securing their personal savings to better respond to health shocks. In case of imminent need, i.e. due to sudden illness or complications, such an approach further enables users to collect contributions from relatives or raise funds from other sources. 

We furthermore elucidated why pregnancy related care is particularly well suited for a savings scheme like the MHW (page 4, line 87): 

Pregnancy and birth-related care seems to be particularly well suited for the use of a MHW as expenses are mostly predictable both in time and amount. However, to our knowledge, there is currently no evidence on the success of application of a MHW specifically for maternal health care.

5. Line 146 Prior to interview ---- move this to data collection, does not fall under sampling, unless if the data were used to decide who could be interviewed.

-> We have moved the information on participants and topics to the data collection section. 

6. The data analysis needs to be explained in more detail for the reader to understand how the themes were arrived at and to illustrate an audit trail underpinning the trustworthiness. I like the way the conceptual framework was weaved into the findings. However, it would be good to mention the sub-themes within the broader categories of the conceptual framework

-> We thank the reviewer for this suggestion to explain the details of our data analysis. We have thus added more details on how the analysis was performed (page 9, line 201): 

Transcripts were analyzed by two researchers (NM, AF) who developed a codebook which was validated by a senior qualitative research expert (SM) (30). Each set of data was analyzed at least twice. Thematic coding was applied by using a partially deductive approach, informed by findings from a previous study, and an inductive coding technique with creation of themes as they emerged from the transcripts (13). Analysts convened regularly to debate differences in coding and interpretation. During the analysis process, when new codes emerged, coders discussed the new findings and transcripts were recoded as the codebook was refined.

In the data analysis section, we now provide more information on themes and subthemes that emerged during the process (page 9, line 217): 

(i) Institutional and policy: Influences of the current health care system and implications of national and local laws and policies. Sub-themes include previous experience with mobile payment schemes, implications by the existing service payment system, potential conflicts with external service providers and legal and technical factors with implications on the MHW.

(ii) Interpersonal and community: Factors by formal and informal social networks and social support systems. Subthemes describe interpersonal and cultural influences on decision-making, influences by applied saving methods and factors related to communication strategies. 

(iii) Individual: Factors and characteristics of the individual that influence behavior, expectations and attitude towards the MHW. Subthemes focus on the minimum prerequisites for access and factors influencing the individuals’ decision-making.

7. Line 170 - gender and distribution - can this be put in a table format? it would be easier to read in a table format.

-> We have created a new figure (Figure 1) to depict the gender distribution of participants (cf. response to comment 4 by reviewer 1). 

8. The discussion should highlight the limitations of the MHW given the low socioeconomic status of most of the pregnant mothers as well as the absence of handsets and the literacy rates. The illiteracy was mitigated well by suggestions of use of illustrations for health education. Or perhaps given these findings this is something that should be tried with a middle-income cohort of pregnant women?

-> We thank the reviewer for this important concern. We entirely agree, that socioeconomic status is potentially reducing access to the MHW and added this important limitation to the corresponding section 8page 19 line 469): 

Although the MHW's primary goal is to reduce barriers to health care services for populations most in need, women from the lower socioeconomic strata, including women who lack basic school education and/or the means to afford a mobile phone or SIM card, might not sign up to the MHW. To minimize health inequities, we strongly recommend more research on prevailing prerequisites of access, leading to a more inclusive design of the MHW.

We also added this point to the conclusion section (page 20, line 483): 

More research is needed to address factors determining access to health care services among pregnant women from lower socioeconomic strata in more detail. 

9. The suggestion of several people using a handset should take into consideration issues of confidentiality. I wish this question had been posed to the participants to hear their take on it. In my experience, many men would not want to share a phone with a woman.

-> We unfortunately didn’t pose this question to the study participants. Results from our previous study among pregnant women and young mothers in the same study area showed that close to 30% of women do not own a personal phone but have regular access to a family phone. We added this finding to the discussion section (page 18, line 418): 

Based on these contradictory but potentially fundamental findings and the prevailing reality of adult illiteracy in the Malagasy population, we strongly suggest applying an image-based knowledge dissemination strategy and alternatively offering verbal announcements and voice calls to convey basic program information. {…} However, it needs to be considered that a single SIM card for several people raises concerns of confidentiality and security of individual savings. We therefore suggest making SIM cards largely available to pregnant women even without owning a phone as they can be inserted and removed easily with any handset, thus mitigating the aforementioned privacy concerns.

We furthermore stressed the fact that all personal data and access to funds are connected to the SIM card which can be removed and kept individually, minimizing the issue of confidentiality. We have further clarified this in the following sentence in the introduction section (page 4, line 95): 

Personal data and funds on the MHW are linked to an individual SIM (subscriber identity module) card which can be used with any type of handset independent of smartphone capability or internet access.

10. This was an interesting study that puts a different spin to mHealth and health insurance, as the women would be in control of the money, not the health insurance.

-> We thank the reviewer for this encouraging comment and the recognition of the ultimate aim of our work, which is to empower vulnerable populations to access essential maternal health care.

---

## [Editor Report · Decision Letter 1]

7 Jan 2020

Facilitators and barriers to the implementation of a Mobile Health Wallet for pregnancy-related health care: a qualitative study of stakeholders’ perceptions in Madagascar

PONE-D-19-28907R1

Dear Dr. Nadine Muller,

We are pleased to inform you that your manuscript has been judged scientifically suitable for publication and will be formally accepted for publication once it complies with all outstanding technical requirements.

With kind regards,

Sharon Mary Brownie

Academic Editor

PLOS ONE
---

## [Editor Report · Acceptance letter]

10 Jan 2020

PONE-D-19-28907R1 

Facilitators and barriers to the implementation of a Mobile Health Wallet for pregnancy-related health care: a qualitative study of stakeholders’ perceptions in Madagascar 

Dear Dr. Muller:

I am pleased to inform you that your manuscript has been deemed suitable for publication in PLOS ONE. Congratulations! Your manuscript is now with our production department. 

With kind regards,

on behalf of

Professor Sharon Mary Brownie 

Academic Editor

PLOS ONE